# Transferable Perturbations of Deep Feature Distributions

**Nathan Inkawhich, Kevin J Liang, Lawrence Carin & Yiran Chen**
Department of Electrical and Computer Engineering
Duke University
{nathan.inkawhich,kevin.liang,lcarin,yiran.chen}@duke.edu

## Abstract

Almost all current adversarial attacks of CNN classifiers rely on information derived from the output layer of the network. This work presents a new adversarial attack based on the modeling and exploitation of class-wise and layer-wise deep feature distributions. We achieve state-of-the-art targeted blackbox transfer-based attack results for undefended ImageNet models. Further, we place a priority on explainability and interpretability of the attacking process. Our methodology affords an analysis of how adversarial attacks change the intermediate feature distributions of CNNs, as well as a measure of layer-wise and class-wise feature distributional separability/entanglement. We also conceptualize a transition from task/data-specific to model-specific features within a CNN architecture that directly impacts the transferability of adversarial examples.

## 1 Introduction

Most recent adversarial attack literature has focused on empirical demonstrations of how classifiers can be fooled by the addition of quasi-imperceptible noise to the input (Szegedy et al., 2014; Goodfellow et al., 2015; Carlini & Wagner, 2017; Moosavi-Dezfooli et al., 2016; Madry et al., 2018; Kurakin et al., 2017). However, adversarial attacks may be leveraged in other constructive ways to provide insights into how deep learning models learn data representations and make decisions. In this work, we propose a new blackbox transfer-based adversarial attack that outperforms state-of-the-art methods for undefended ImageNet classifiers. Importantly, this work provides a broad exploration into how different Deep Neural Network (DNN) models build feature representations and conceptualize classes.

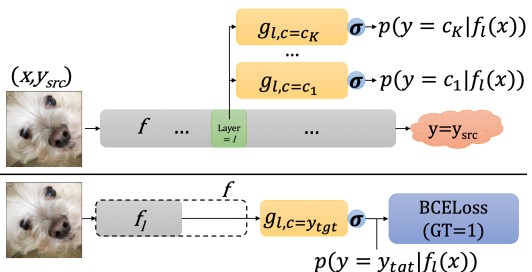

Figure 1: (top) Given a pre-trained whitebox model $f$, we capture the layer-wise and class-wise feature distributions with binary neural networks $g_{l,c}$, aiming to model the probability that the layer $l$ features extracted from input $x$ are from the class $c$ feature distribution (i.e. $p(y = c|f_l(x))$). (bottom) Forward pass for *FDA* targeted attack.

The new attack methodology, which we call the Feature Distribution Attack (*FDA*), leverages class-wise and layer-wise deep feature *distributions* of a substitute DNN to generate adversarial examples that are highly transferable to a blackbox target DNN.

One perspective on adversarial attacks is that adversarial noise is a *direction* in which to "move" the natural data. In standard attacks which directly use the classification output, the noise points in the direction of the nearest decision boundary at the classification layer (Tramèr et al., 2017). In this work, our crafted noise points in a direction that makes the data "look like" a sample of another class in *intermediate feature space*. Intuitively, if we can alter the representation in a layer whose features are representative of the data for the given task, but not specific to the model, the adversarial example may transfer better (to unobserved architectures) than attacks derived from logit-layer information.

Figure 1(top) illustrates the feature distribution modeling of a DNN, which is the core mechanism of the attack. $f$ is a pre-trained substitute whitebox model to which we have full access. The true target blackbox model is not shown, but we only assume limited query access and that it has been

trained on ImageNet-1k (Deng et al., 2009). Adversarial examples are then generated on the white-box model and transferred to the blackbox model. The novelty of the attack comes from the explicit use of class-wise and layer-wise feature distributions. In Figure 1(top), an auxiliary Neural Network (NN) $g_{l,c}$ learns $p(y = c|f_l(x))$, which is the probability that the layer $l$ features of the whitebox model, extracted from input image $x$, belong to class $c$. The attack uses these learned distributions to generate targeted (or untargeted) adversarial examples by maximizing (or minimizing) the probability that the adversarial example is from a particular class's feature distribution (Figure 1(bottom)). We also use these learned distributions to analyze layer-wise and model-wise transfer properties, and to monitor how perturbations of the input change feature space representations. Thus, we gain insights on how feature distributions evolve with layer depth and architecture.

## 2 RELATED WORK

In blackbox attacks (Narodytska & Kasiviswanathan, 2017; Su et al., 2017; Papernot et al., 2017; Tramèr et al., 2017; Inkawhich et al., 2019; Dong et al., 2018; Zhou et al., 2018), knowledge of the target model is limited. In this work, the target model is blackbox in the sense that we do not have access to its gradients and make no assumptions about its architecture (Madry et al., 2018; Cheng et al., 2019). A popular blackbox technique is transfer-based attacks, in which adversarial examples are constructed on the attackers' own whitebox model and *transferred* to the target model. Papernot et al. (2016; 2017) develop special methods for training the attackers' whitebox model to approximate the target model's decision boundaries. In this work, we only use models that have been trained under standard configurations for ImageNet-1k (Deng et al., 2009). Tramèr et al. (2018) and Liu et al. (2017) bolster transferability by generating adversarial examples from an ensemble of whitebox models, which helps the noise not overfit a single model architecture. Our methods also discourage overfitting of the generating architecture, but we instead leverage feature space perturbations at the appropriate layer. In the 2017 NeurIPS blackbox attack competition (Kurakin et al., 2018), the winning method (Dong et al., 2018) used momentum in the optimization step, which helped to speed up the convergence rate and de-noise the gradient directions as to not be overly specific to the generating architecture. We also use this approach. Finally, Tramèr et al. (2017) analyze why transferability occurs and find that well-trained models have similar decision boundary structures. We also analyze transferability, but in the context of how adversarial examples change a model's internal representations, rather than only making observations at the output layer.

While all of the above methods generate adversarial examples using information from the classification layer of the model, there have been a few recent works delving into the feature space of DNNs for both attacks and defenses. Sabour et al. (2016) show that in whitebox settings, samples can be moved very close together while maintaining their original image-domain representations. Zhou et al. (2018) regularize standard untargeted attack objectives to maximize perturbations of (all) intermediate feature maps and increase transferability. However, their primary objective is untargeted and still based on classification output information. Also, the authors do not consider which layers are affected and how the regularization alters the intermediate representations. Inkawhich et al. (2019) show that driving a source sample's feature representation towards a target sample's representation at particular layers in deep feature space is an effective method of targeted transfer attack. However, the method is targeted only and relies on the selection of a single (carefully selected) sample of the target class. Also, the attack success rate on ImageNet was empirically low. This work describes a more robust attack, with significantly better performance on ImageNet, and provides a more detailed analysis of layer-wise transfer properties. For adversarial defenses, Xie et al. (2019), Frosst et al. (2019), and Lin et al. (2019) consider the effects of adversarial perturbations in feature space but do not perform a layer-wise analysis of how the internal representations are affected.

## 3 ATTACK METHODOLOGY

We assume to have a set of training data and a pre-trained model $f$ from the same task as the target blackbox model (i.e. the ImageNet-1k training set and a pre-trained ImageNet model). To model the feature distributions for $f$, we identify a set of classes $\mathcal{C} = \{c_1, ..., c_K\}$ and a set of layers $\mathcal{L} = \{l_1, ..., l_N\}$ that we are keen to probe. For each layer in $\mathcal{L}$, we train a small, binary, one-versus-all classifier $g$ for each of the classes in $\mathcal{C}$, as shown in Figure 1 (top). Each binary classifier is given a unique set of parameters, and referred to as an auxiliary model $g_{l,c}$. The output of an auxiliary model represents the probability that the input feature map is from a specific class $c \in \mathcal{C}$. Thus, we

say that $g_{l,c}(f_l(x))$ outputs $p(y = c|f_l(x))$, where $f_l(x)$ is the layer $l$ feature map of the pre-trained model $f$ given input image $x$.

Once trained, we may leverage the learned feature distributions to create both targeted and untargeted adversarial examples. Here, we focus mostly on targeted attacks, which are considered a harder problem (especially in the blackbox transfer case when we do not have access to the target model's gradients) (Kurakin et al., 2018; Sharma et al., 2018). Discussion of untargeted attacks is left to Appendix C. Recall, the goal of a targeted attack is to generate an adversarial noise $\delta$ that when added to a clean sample $x$ of class $y_{src}$, the classification result of $x + \delta$ is a chosen class $y_{tgt}$. The key intuition for our targeted methods is that if a sample has features consistent with the feature distribution of class $c$ at some layer of intermediate feature space, then it will likely be classified as class $c$. Although not shown in the objective functions for simplicity, for all attacks the adversarial noise $\delta$ is constrained by an $\ell_p$ norm (i.e. $||\delta||_p \le \epsilon$), and the choice of layer $l$ and target class label $y_{tgt}$ are chosen prior to optimization.

***FDA*** We propose three targeted attack variants. The most straightforward variant, called *FDA*, finds a perturbation $\delta$ of the "clean" input image $x$ that maximizes the probability that the layer $l$ features are from the target class $y_{tgt}$ distribution:

$$\max_{\delta} p(y = y_{tgt}|f_l(x + \delta)). \tag{1}$$

We stress that unlike standard attacks that use output layer information to directly cross decision boundaries of the whitebox, our *FDA* objective leverages *intermediate feature distributions* which do not implicitly describe these exact boundaries.

***FDA+ms*** In addition to maximizing the probability that the layer $l$ features are from the target class distribution, the *FDA+ms* variant also considers minimizing the probability that the layer $l$ features are from the source class $y_{src}$ distribution ($ms$ = minimize source):

$$\max_{\delta} \lambda p(y = y_{tgt}|f_l(x + \delta)) - (1 - \lambda)p(y = y_{src}|f_l(x + \delta)). \tag{2}$$

Here, $\lambda \in (0, 1)$ weights the contribution of both terms and is a fixed positive value.

***FDA+fd*** Similarly, the *FDA+fd* variant maximizes the probability that the layer $l$ features are from the target class distribution while also maximizing the distance of the perturbed features from the original features ($fd$ = feature-disruption):

$$\max_{\delta} p(y = y_{tgt}|f_l(x + \delta)) + \eta \frac{\|f_l(x + \delta) - f_l(x)\|_2}{\|f_l(x)\|_2}. \tag{3}$$

In other words, the feature-disruption term, with a fixed $\eta \in \mathbb{R}_+$, prioritizes making the layer $l$ features of the perturbed sample maximally different from the original sample.

The additional terms in *FDA+ms* and *FDA+fd* encourage the adversarial sample to move far away from the starting point, which may intuitively help in generating (targeted) adversarial examples. Also, notice that *FDA+ms* requires the modeling of both the source and target class distributions, whereas the others only require the modeling of the target class distribution.

**Optimization Procedure.** The trained auxiliary models afford a way to construct a fully differentiable path for gradient-based optimization of the objective functions. Specifically, to compute *FDA* adversarial noise from layer $l$, we first build a composite model using the truncated whitebox model $f_l$ and the corresponding layer's auxiliary model $g_{l,c=y_{tgt}}$ for the target class $y_{tgt}$, as shown in Figure 1(bottom). The loss is calculated as the Binary Cross Entropy (BCELoss) between the predicted $p(y = y_{tgt}|f_l(x))$ and 1. Thus, we perturb the input image in the direction that will minimize the loss, in turn maximizing $p(y = y_{tgt}|f_l(x))$. For optimization, we employ iterative gradient descent with momentum, as the inclusion of a momentum term in adversarial attacks has proven effective (Inkawhich et al., 2019; Dong et al., 2018). See Appendix D for more details.

## 4 EXPERIMENTAL SETUP

**ImageNet models.** For evaluation we use popular CNN architectures designed for the ImageNet-1k (Deng et al., 2009) classification task: VGG-19 with batch-normalization (VGG19) (Simonyan & Zisserman, 2015), DenseNet-121 (DN121) (Huang et al., 2017), and ResNet-50 (RN50) (He et al.,

2016). All models are pre-trained and found in the PyTorch Model Zoo. Note, our methods are in no way specific to these particular models/architectures. We also emphasize transfers across different architectures rather than showing results between models from the same family.

**Layer decoding scheme**. Given a pre-trained model, we must choose a set of layers $\mathcal{L}$ to probe. For each model we subsample the layers such that we probe across the depth. For notation we use relative layer numbers, so layer 0 of DN121 ($DN121_{l=0}$) is near the input layer and $DN121_{l=12}$ is closer to the classification layer. For all models, the deepest layer probed is the logit layer. Appendix A decodes the notation for each model.

**Auxiliary model training.** We must also choose a set of classes $\mathcal{C}$ that we are interested in modeling. Recall, the number of auxiliary models required for a given base model is the number of layers probed multiplied by the number of classes we are interested in modeling. Attempting to model the feature distributions for all 1000 ImageNet classes for each layer is expensive, so we instead choose to run the majority of tests with a set of 10 randomly chosen classes (which are meant to be representative of the entire dataset): 24:"grey-owl", 99:"goose", 245:"bulldog", 344:"hippo", 471:"cannon", 555:"fire-truck", 661:"Model-T", 701:"parachute", 802:"snowmobile", 919:"street-sign". Thus, for each layer of each model we train 10 auxiliary classifiers, one for each class. After identifying high performing attack settings, we then produce results for all 1000 classes.

The architecture of all auxiliary models is the same, regardless of model, layer, or class. Each is a 2-hidden layer NN with a single output unit. There are 200 neurons in each hidden layer and the number of input units matches the size of the input feature map. To train the auxiliary models, unbiased batches from the whole ImageNet-1k training set are pushed through the truncated pre-trained model ($f_l$), and the extracted features are used to train the auxiliary model parameters.

**Experimental procedure.** Since we have three pre-trained models, there are 6 blackbox transfer scenarios to evaluate (no self-transfers). We use the ImageNet-1K validation set as the test dataset. Because *FDA+ms* requires both the source and target class distributions, for the targeted attack evaluations we only use source samples from the 10 trained classes, and for each sample, target each of the other 9 classes. For baseline attacks, we use targeted random-start Projected Gradient Descent (tpgd) (Madry et al., 2018; Kurakin et al., 2018), targeted NeurIPS2017 competition winning momentum iterative method (tmim) (Dong et al., 2018), and the Activation Attack (AA) (Inkawhich et al., 2019). Further, all targeted adversarial examples are constrained by $\ell_\infty$ $\epsilon = 16/255$ as described in (Dong et al., 2018; Kurakin et al., 2018). As experimentally found, $\lambda = 0.8$ in (2) and $\eta = 1\text{e-}6$ in (3). Finally, as measured over the initially correctly classified subset of the test dataset (by both the whitebox and blackbox models), attack success is captured in two metrics. *Error* is the percentage of examples that the blackbox misclassifies and *Targeted Success Rate* (tSuc) is the percentage of examples that the blackbox misclassifies as the target label.

## 5 EMPIRICAL RESULTS

### 5.1 10-CLASS IMAGENET RESULTS

The primary axis of interest is how attack success rate varies with the layer depth from which the feature distributions are attacked. Figure 2 shows the transfer results between all pairs of whitebox and blackbox models. Each plot shows a metric of attack success versus relative layer depth of the generated attack. The notation DN121 → VGG19 indicates adversarial examples were generated with a DN121 whitebox model and transferred to a VGG19 blackbox model.

Similar to Inkawhich et al. (2019), transferability trends for *FDAs* from a given whitebox model appear blackbox model agnostic (e.g. the shape of the curves from DN121 → RN50 are the same as DN121 → VGG19). This is a positive property, as once the optimal transfer layer for a whitebox model is found, evidence shows it will be the same for any blackbox model architecture. Further, the most powerful transfers come from perturbations of intermediate features, rather than perturbations of classification layer information. Another global trend is that in tSuc, *FDA+fd* performs best, *FDA* performs worst, *FDA+ms* is in-between, and all *FDAs* significantly outperform the other baselines. In the error metric, *FDA+fd* is best in early layers, *FDA+ms* is best in later layers, and *FDA* routinely under-performs the AA baseline. Although all attacks are targeted, it is relevant to report error as it is still an indication of attack strength. Also, it is clearly beneficial for the targeted attack objective to include a term that encourages the adversarial example to move far away from its starting place (*FDA+ms* & *FDA+fd*) in addition to moving toward the target region.

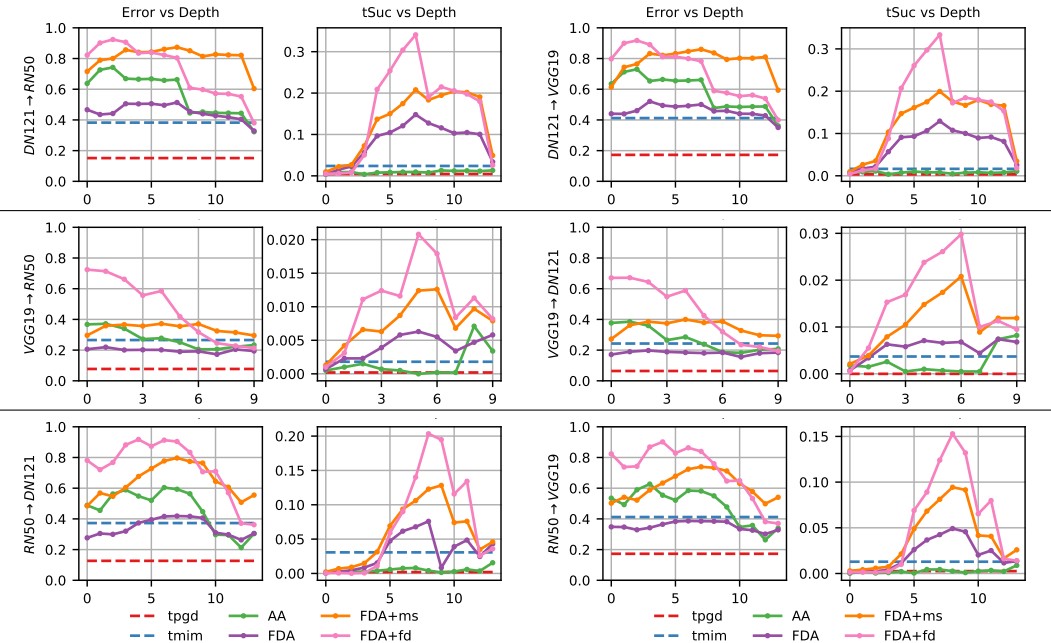

Figure 2: Targeted adversarial attack transfer results. The x-axis of each plot is the relative layer depth at which the adversarial example was generated from. Each row is a different whitebox model.

We now compare performance across whitebox models. For a DN121 whitebox, *FDA+fd* from $DN121_{l=7}$ is the optimal targeted attack with an average tSuc of 34%. For DN121 $\rightarrow$ RN50, this attack outperforms the best baseline by 14% and 32% in error and tSuc, respectively. For a VGG19 whitebox, *FDA+fd* from $VGG19_{l=5}$ is the optimal targeted attack with an average tSuc of 2.3%. For VGG19 $\rightarrow$ RN50, this attack outperforms the best baseline by 15% and 2% in error and tSuc, respectively. For a RN50 whitebox, *FDA+fd* from $RN50_{l=8}$ is the optimal targeted attack with an average tSuc of 18%. For RN50 $\rightarrow$ DN121, this attack outperforms the best baseline by 27% and 17% in error and tSuc, respectively.

## 5.2 1000-CLASS IMAGENET RESULTS

Table 1: Transferability rates for 1000-class targeted attack tests using optimal layers.

| attack | DN121 $\rightarrow$ VGG19 | | DN121 $\rightarrow$ RN50 | | RN50 $\rightarrow$ DN121 | | RN50 $\rightarrow$ VGG19 | |
|---|---|---|---|---|---|---|---|---|
| | error | tSuc | error | tSuc | error | tSuc | error | tSuc |
| tpgd | 23.1 | 0.3 | 21.4 | 0.6 | 20.2 | 0.5 | 22.4 | 0.3 |
| tmim | 48.6 | 1.4 | 45.5 | 2.2 | 44.3 | 2.9 | 46.7 | 1.3 |
| FDA | 64.9 | 15.5 | 64.3 | 18.1 | 56.4 | 12.6 | 54.6 | 6.9 |
| FDA+ms | **91.9** | 21.7 | **91.9** | 23.4 | **87.3** | 15.9 | **85.3** | 10.2 |
| FDA+fd | 81.2 | **29.0** | 81.7 | **30.9** | 82.6 | **24.3** | 78.9 | **15.9** |

Recall, due to the computational complexity of training one auxiliary model per class per layer per model, we ran the previous experiments using 10 randomly sampled ImageNet-1k classes. In reality, this may be a realistic attack scenario because an adversary would likely only be interested in attacking certain source-target pairs. However, to show that the 10 chosen classes are not special, and the previously identified optimal transfer layers are still valid, we train all 1000 class auxiliary models for $DN121_{l=7}$ and $RN50_{l=8}$. We exclude VGG19 because of its inferior performance in previous tests. Table 1 shows results for the four transfer scenarios. Attack success rates are all averaged over four random 10k splits and the standard deviation of all measurements is less than 1%. In these tests, for each source sample, a random target class is chosen. As expected, the 1000-class results closely match the previously reported 10-class results.

## 6 ANALYSIS OF TRANSFER PROPERTIES

We now investigate why a given layer and/or whitebox model is better for creating transferable adversarial examples. We also explore the hypothesis that the layer-wise transfer properties of a

DNN implicate the transition of intermediate features from *task/data-specific* to *model-specific*. Intuitively, early layers of DNNs trained for classification may be working to optimally construct a task/data-specific feature set (Zeiler & Fergus, 2014; Yosinski et al., 2014). However, once the necessary feature hierarchy is built to model the data, further layers may perform extra processing to best suit the classification functionality of the model. This additional processing may be what makes the features model-specific. *We posit that the peak of the tSuc curve for a given transfer directly encodes the inflection point from task/data-specific to model-specific features in the whitebox model.* Instinctively, to achieve targeted attack success, the layer at which the attacks are generated must have captured the concepts of the classes for the general task of classification, without being overly specific to the architecture. Thus, layers prior to the inflection point may not have solidified the class concepts, whereas layers after the inflection point may have established the class concepts and are further processing them for the model output. This may also be considered an extension of the convergent learning theory of Li et al. (2016) and general-to-specific theory of Yosinski et al. (2014).

## 6.1 INTERMEDIATE DISRUPTION

One way to measure why and how adversarial attacks work is to observe how the intermediate representations change as a result of perturbations to the input. Our trained auxiliary models afford a novel way to monitor the effects of such perturbations in deep feature space. To measure how much a layer's features have changed as a result of a (targeted) adversarial perturbation, we define layerwise *disruption* as the difference between the target class probability before and after perturbation, as measured in layer $l$ of model $f$: disruption $= p(y = y_{tgt}|f_l(x + \delta)) - p(y = y_{tgt}|f_l(x))$.

Figure 3 shows the average disruption caused in each transfer scenario, using both logit-based (tmim) and feature-based (*FDA+fd*) adversarial attacks. Each row plots the disruption versus layer depth from a single whitebox model to each blackbox model (e.g. the top row results from DN121 → VGG19 and DN121 → RN50 transfers). Each line represents the average disruption caused by some adversarial attack, where all FDAs are *FDA+fd*. The first column of plots shows the impact of each attack on the whitebox model's feature distributions while the second and third columns shows impacts on the blackbox models' feature distributions.

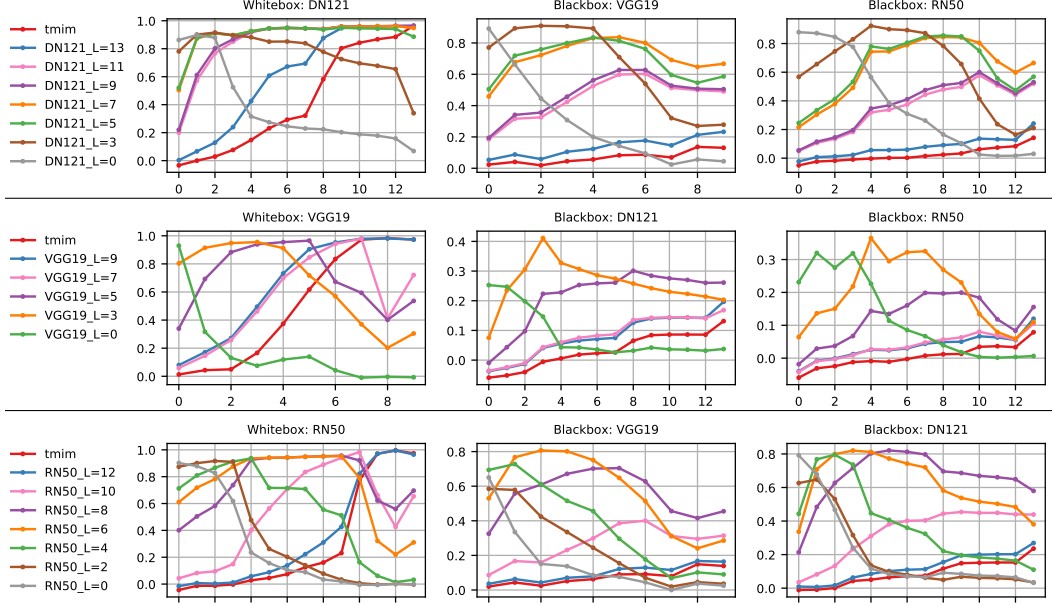

Figure 3: Disruption versus layer depth for all transfer scenarios. Each row uses a different whitebox model. Each line is a different attack, where all FDAs are *FDA+fd*.

It appears FDAs generated from early layers (e.g. $DN121_{l=0}$, $VGG19_{l=0}$, $RN50_{l=0}$) disrupt features the most in early layers and less so in deeper layers. Therefore, a sample resembling class $y_{tgt}$ in an early layer does not mean it will ultimately be classified as $y_{tgt}$. However, recall from Figure 2 that attacks from early layers create very powerful untargeted adversarial examples (error). This indicates that early layer perturbations are amplified as they proceed through the model (Lin

et al., 2019), just not in a class-specific manner. Next, as expected, attacks that use information from the last layers of a whitebox model (e.g. *tmim*, $DN121_{l=13}$, $VGG19_{l=9}$, $RN50_{l=12}$) create the largest disruption in the last layers of the whitebox, but not necessarily at the last layers of the blackbox models. However, the optimal transfer attacks ($DN121_{l=7}$, $VGG19_{l=5}$, $RN50_{l=8}$) have high disruption all throughout the models, not just at the last layer. This is further evidence that perturbations of classification-layer features are overly model-specific and perturbations of optimal transfer-layer features are more specific to the data/task. Finally, notice that the maximum disruption caused in any blackbox model layer from VGG19 whitebox transfers is around 40% (row 2). For the DN121 and RN50 whitebox models, the maximum disruption is around 80%. This may explain VGG19 being an inferior whitebox model to transfer from, as the perturbation of intermediate VGG features does not in-turn cause significant disruption of blackbox model features.

## 6.2 AUXILIARY MODEL CORRELATION WITH FULL MODEL

Another point of analysis is to investigate the correlation/discrepancy between the auxiliary models at a given layer and the output of the whitebox model. This may also indicate a transition from task/data-specific to model-specific features. We discuss *discrepancy* as an indication of how different the auxiliary model outputs are from the whitebox model outputs. Then *correlation* is the inverse of discrepancy so that when the auxiliary model outputs align well with the whitebox model outputs, the discrepancy is low and correlation is high. To evaluate discrepancy at a given layer $l$ for input $x$, we aggregate the logit values (i.e. pre-sigmoid/softmax) for each class in $\mathcal{C}$, as measured by the auxiliary models $g_{l,c}$ and the whitebox model $f$, into separate vectors. Then, a softmax (smax) operation is performed on each vector to establish two proper probability distributions over the classes in $\mathcal{C}$. Discrepancy is then defined as the Kullback-Leibler divergence ($D_{\mathrm{KL}}$) between the two

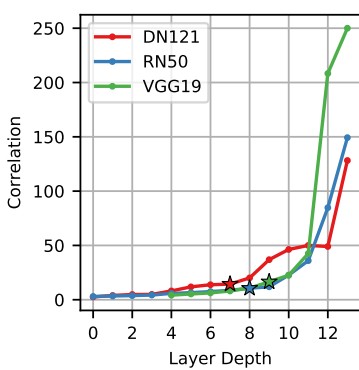

Figure 4: Correlation of a layer's auxiliary models with the whitebox model output.

distributions, or $\mathrm{discrepancy} = D_{\mathrm{KL}}\big(\mathrm{smax}\big(\big[g_{l,c}(f_l(x))\big]_{\forall c\in\mathcal{C}}\big) \,\|\, \mathrm{smax}\big(\big[f(x)[c]\big]_{\forall c\in\mathcal{C}}\big)\big)$. Here, $f(x)[c]$ is the class $c$ logit value from the whitebox model $f$ given input $x$. Figure 4 shows the layer-wise auxiliary model correlations with the whitebox model outputs as measured from the average discrepancy over 500 input samples of classes in $\mathcal{C}$.

Note, the shapes of the curves are more informative than the actual values. Also, VGG19 layers have been shifted in notation by +4 so that layer depth 13 is the logit layer of each model. As expected, the auxiliary models in early layers have little correlation with the model output, while auxiliary models in later layers have high correlation with the model output. Importantly, the optimal transfer layers ($\star$) mark a transition in the trendlines after which correlation increases sharply. This effect may directly explain why layers after the optimal layer are suboptimal, because the auxiliary models become highly correlated with the model output and begin to overfit the architecture. Since the auxiliary models are not highly-correlated with the output layer at the optimal-transfer-layers, we may surmise that the captured features are still mostly task/data specific.

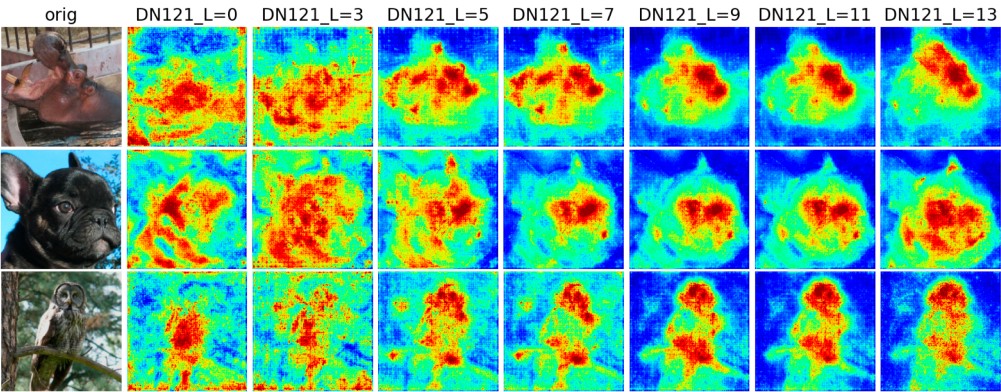

Figure 5: Saliency maps of auxiliary models on several interesting inputs across model depth.

### 6.3 AUXILIARY MODEL SALIENCY

For a more qualitative analysis, we may inspect the auxiliary model saliency maps. Given an image of class $y_{src}$, we visualize in Figure 5 what is salient to the $y_{src}$ auxiliary models at several DN121 layer depths using SmoothGrad (Smilkov et al., 2017) (see Appendix E for additional saliency examples for RN50). Notice, an observable transition occurs at the high-performing transfer layers from Figure 2 ($DN121_{l=5,7}$). The salient regions move from large areas around the whole image (e.g. $DN121_{l=0,3}$) to specific regions that are also salient in the classification layer ($DN121_{l=13}$). The saliency and correlation transitions together show that the well-transferring layers have learned similar salient features as the classification layer while not being overly correlated with the model output. Therefore, perturbations focused on these salient regions significantly impact the final classification without being too specific to the generating architecture.

### 6.4 CLASS DISTRIBUTION SEPARABILITY

Finally, our trained auxiliary models afford new ways of measuring class-wise feature entanglement/separability for the purpose of explaining transfer performance. We adopt the definition of *entanglement* from Frosst et al. (2019) which states that highly entangled features have a "lack of separation of class manifolds in representation space," and define *separability* as the inverse of entanglement. One way to measure the separability between class distributions in a layer using the auxiliary models is to gauge how far a sample has to "move" to enter a region of high class confidence. We define *intra-class distance* as the distance a sample has to move to enter a high-confidence region of its source class's distribution. Similarly, we define *inter-class distance* as the distance a sample has to move to enter a high-confidence region of a target class distribution, where $y_{src} \neq y_{tgt}$. Then, separability in a layer is

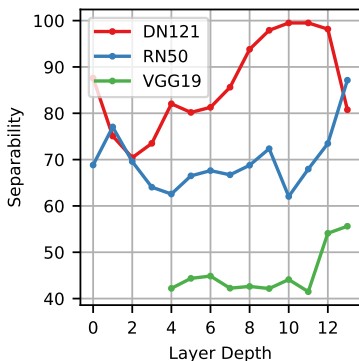

Figure 6: Class separability versus layer depth for each whitebox.

the difference between average inter-class and intra-class distances. In practice, for a given sample, given model, chosen target class, and chosen layer, we iteratively perturb the sample using *FDA* with a small step size (e.g. 5e-5) until the confidence of the target distribution auxiliary network is over a threshold (e.g. 99.9%). The number of perturbation steps it takes to reach the confidence threshold encodes the distance of the sample to the targeted class's distribution.

Results for the three whitebox models are shown in Figure 6, where the vertical axis is the separability in units of perturbation steps, and the horizontal axis is the layer-depth of each model. We see that there is some separability in all layers, for all models, indicating that even features very close to the input layer are somewhat class-specific. Further, VGG19's features are much less separable than DN121 and RN50, indicating why VGG19 may have performed much worse as a whitebox model in the transferability tests. In the same vein, DN121 has generally the most separated features which further indicates why it may be a superior whitebox model. Intuitively, if a model/layer has highly class-separable feature distributions, FDA attacks may be more transferable because there is less ambiguity between the target class's distribution and other class distributions during generation.

## 7 CONCLUSIONS

We present a new targeted blackbox transfer-based adversarial attack methodology that achieves state-of-the-art success rates for ImageNet classifiers. The presented attacks leverage learned class-wise and layer-wise intermediate feature *distributions* of modern DNNs. Critically, the depth at which features are perturbed has a large impact on the transferability of those perturbations, which may be linked to the transition from task/data-specific to model-specific features in an architecture. We further leverage the learned feature distributions to measure the entanglement/separability of class manifolds in the representation space and the correlations of the intermediate feature distributions with the model output. Interestingly, we find the optimal attack transfer layers have feature distributions that are class-specific and highly-separable, but are not overly-correlated with the whitebox model output. We also find that highly transferable attacks induce large disruptions in the intermediate feature space of the blackbox models.

ACKNOWLEDGMENTS

The research was supported in part by AFRL (FA8750-18-2-0057), DARPA, DOE, NIH, NSF and ONR.

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

APPENDIX

A. LAYER DECODING

Table 2: Whitebox Model Layer Decoding Table

| Layer | DenseNet-121 | VGG19bn | ResNet-50 |
|---|---|---|---|
| 0 | 6,2 | 512 | 3,1 |
| 1 | 6,10 | 512 | 3,2 |
| 2 | 6,12 | 512 | 3,3 |
| 3 | 6,12,2 | 512 | 3,4 |
| 4 | 6,12,14 | 512 | 3,4,1 |
| 5 | 6,12,20 | 512 | 3,4,2 |
| 6 | 6,12,22 | 512 | 3,4,3 |
| 7 | 6,12,24 | 512 | 3,4,4 |
| 8 | 6,12,24,2 | FC2 | 3,4,5 |
| 9 | 6,12,24,8 | FC3 | 3,4,6 |
| 10 | 6,12,24,12 | - | 3,4,6,1 |
| 11 | 6,12,24,14 | - | 3,4,6,2 |
| 12 | 6,12,24,16 | - | 3,4,6,3 |
| 13 | 6,12,24,16,FC | - | 3,4,6,3,FC |

Table 2 is the layer number look-up-table that corresponds to the layer notation used in the paper. DenseNet-121 (DN121), VGG19bn (VGG), and ResNet-50 (RN50) appear because they are the model architectures used for the main results. The DN121 notation follows the implementation here: `https://github.com/pytorch/vision/blob/master/torchvision/models/densenet.py`. In english, layer 0 shows that the output of the truncated model comes from the $2^{nd}$ denseblock of the $2^{nd}$ denselayer. Layer 11 means the output of the truncated model comes from the $14^{th}$ denseblock in the $4^{th}$ denselayer. Layer 13 indicates the output comes from the final FC layer of the model.

The VGG model does not have denseblocks or dense layers so we use another notation. In the implementation at `https://github.com/pytorch/vision/blob/master/torchvision/models/vgg.py`, the VGG19bn model is constructed from the layer array: $[64, 64,' M', 128, 128,' M', 256, 256, 256, 256,' M', 512, 512, 512, 512,' M', 512, 512, 512, 512,$ $'M', FC1, FC2, FC3]$, and we follow this convention in the table. In the array, each number corresponds to a convolutional layer with that number of filters, the M's represent max-pooling layers, and the FCs represent the linear layers at the end of the model. Notice, in these tests we do not consider the first 11 layers of VGG19 as they were shown to have very little impact on classification when perturbed.

The RN50 notation follows the implementation here: `https://github.com/pytorch/vision/blob/master/torchvision/models/resnet.py`. As designed, the model has 4 layer groups with [3,4,6,3] Bottlenecks in each, respectively. Thus, layer 0 means the output of the truncated model comes from the $1^{st}$ Bottleneck of layer group 2. Layer 12 means the output comes from the $3^{rd}$ Bottleneck of layer group 4, and layer 13 means the output comes from the final FC layer (i.e. output layer) of the model.

## B. FULL TARGETED TRANSFER RESULTS

Figure 7 shows the full targeted attack transfer results from which the Figure 2 were extracted. These full results include two additional metrics of attack success. *Untargeted Transfer Rate* (uTR) is the rate at which examples that fool the whitebox also fool the blackbox (encodes likelihood of misclassification). *Targeted Transfer Rate* (tTR) is the rate at which successful targeted examples on the whitebox are also successful targeted examples on the blackbox (encodes likelihood of targeted misclassification). Error and tSuc are described in Section 4.

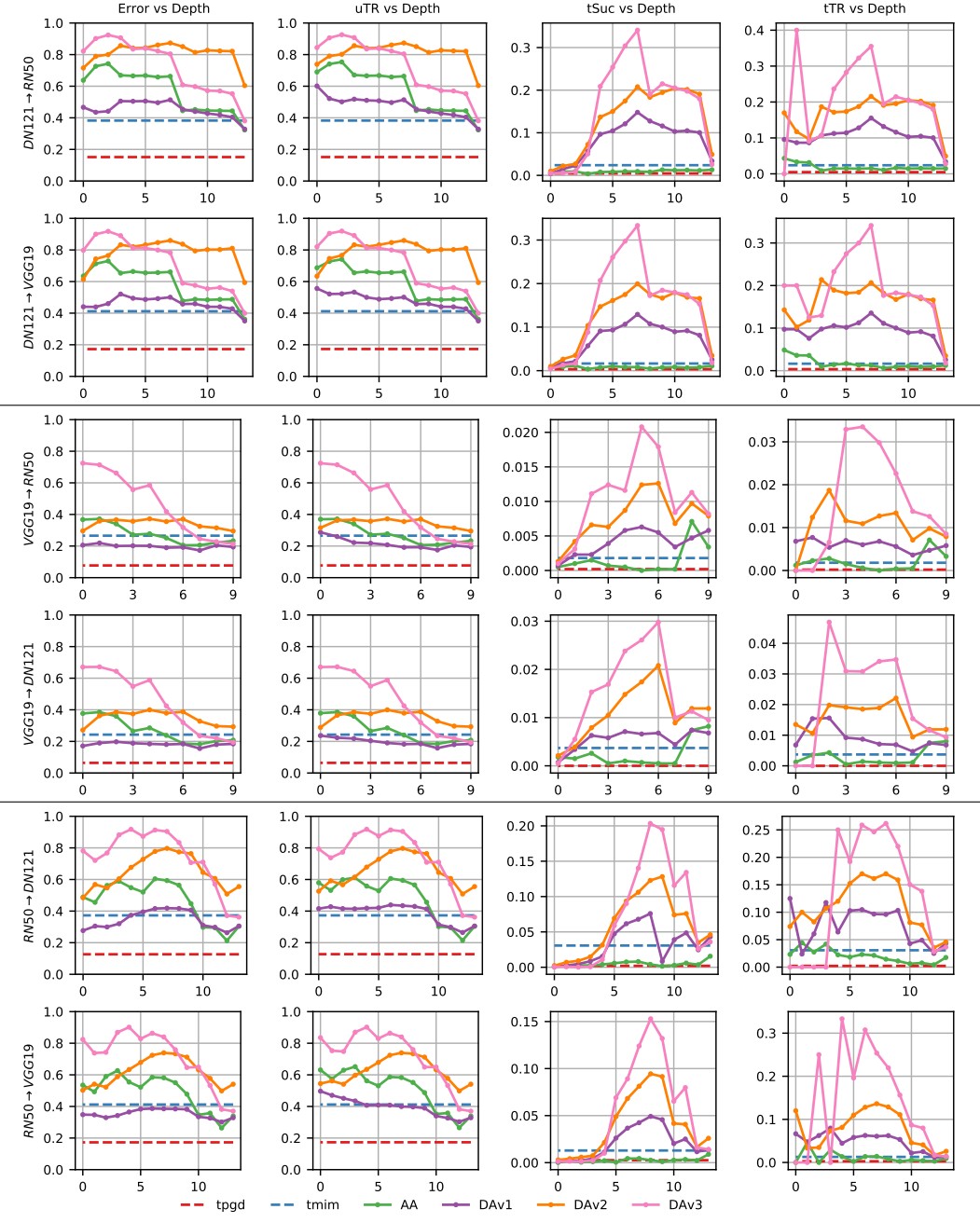

Figure 7: Full targeted adversarial attack transfer results. Each row is a unique transfer scenario and each column is a different attack success metric. The x-axis of each plot is the layer depth at which the adversarial example was generated from. Note, top two rows are transfers from DN121 whitebox model, middle two rows are from VGG19 whitebox model, and bottow two rows are from RN50 whitebox.

C. UNTARGETED FEATURE DISTRIBUTION ATTACKS

The goal of an untargeted attack is to generate an adversarial noise $\delta$ that when added to a clean sample $x$ of class $y_{src}$, the classification result of $x + \delta$ is not $y_{src}$. The key intuition for feature distribution-based untargeted attacks is that if a sample's features are made to be outside of the feature distribution of class $y_{src}$ at some layer of intermediate feature space, then it will likely not be classified as $y_{src}$.

**uFDA**   The first untargeted attack variant is *uFDA* which is described as

$$\min_{\delta} p(y = y_{src} | f_l(x + \delta)).$$

*uFDA* minimizes the probability that the layer $l$ features of the perturbed sample $x + \delta$ are from the source class $y_{src}$ distribution. Unlike the targeted samples which drive towards high confidence regions of a target class feature distribution, this objective drives the sample towards low confidence regions of the source class feature distribution.

**uFDA+fd**   The second untargeted variant *uFDA+fd* is described as

$$\min_{\delta} p_l(y = y_{src} | f_l(x + \delta)) - \eta \frac{\| f_l(x + \delta) - f_l(x) \|_2}{\| f_l(x) \|_2}.$$

*uFDA+fd* also carries a feature disruption term so that the objective drives the perturbed sample towards low confidence regions of the source class feature distribution and maximal distance from the original sample's feature representation.

**fd-only**   The final untargeted attack *fd-only* is described as

$$\max_{\delta} \frac{\| f_l(x + \delta) - f_l(x) \|_2}{\| f_l(x) \|_2}.$$

Notice, *fd-only* is simply the feature disruption term and is a reasonable standalone untargeted attack objective because making features maximally different may intuitively cause misclassification.

To test attack success, we generate untargeted adversarial examples from both DN121 and RN50 whiteboxes and test transfers to a VGG19 blackbox model. It is common to evaluate untargeted attacks with a tighter noise constraint (Kurakin et al., 2018; Dong et al., 2018) as the task is simpler, so in these tests we use $\ell_\infty$ $\epsilon = 4/255$ and $\epsilon = 8/255$ (rather than $\epsilon = 16/255$ used for targeted tests). For baselines, we use the Madry et al. (2018) random start PGD attack (*upgd*) and the Dong et al. (2018) competition winning momentum iterative attack (*umim*). Similar to the layer-wise targeted evaluations, each "clean" source sample belongs to the same set of 10 previously modeled classes. Figure 8 shows the error rates versus layer depth for the attacks.

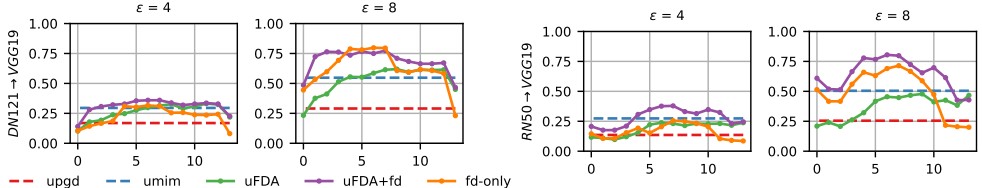

Figure 8: Error versus layer depth plots caused by untargeted adversarial attacks for DN121 $\rightarrow$ VGG19 and RN50 $\rightarrow$ VGG19 transfer scenarios at two different attack strengths $\epsilon = 4, 8$.

As expected, the error rate increases with epsilon and the layer depth at which feature-based attacks are generated from has a large impact on attack success rate. In general, *uFDA+fd* is the top performer, followed by *fd-only*, then *uFDA*. However, *uFDA* often under-performs the *umim* baseline, further indicating that for adversarial attacks in feature space it is beneficial to include a term that prioritizes feature disruption (e.g. *uFDA+fd* & *fd-only*).

On average across models, at $\epsilon = 4/255$, the optimal layer *uFDA+fd* has an untargeted error rate of 37%, which is 9% higher than the best baseline. At $\epsilon = 8/255$, the optimal layer *uFDA+fd* has an untargeted error rate of 79%, which is 27% higher than the best baseline. Also, both whitebox models perform similarly in terms of attack success rate, however the performance of *fd-only* varies

between the two (especially at $\epsilon = 8/255$). Surprisingly, *fd-only* which simply disrupts the original feature map is the optimal attack for the DN121 whitebox (by a small margin). Finally, note that the optimal transfer layers from the targeted attacks (i.e. $DN121_{l=7}$ and $RN50_{l=8}$) are also high performing layers for the untargeted attacks.

## D. ADVERSARIAL EXAMPLE GENERATION PROCESS

Recall, because the auxiliary models are NNs, the optimization objectives described for both the targeted and untargeted attacks can be solved with an iterative gradient descent procedure. For any version of the FDA attacks, we first build a "composite" model which includes the truncated whitebox model $f_l$ and the appropriate auxiliary model $g_{l,c}$, as shown in Figure 1(bottom). An attack loss function $L_{FDA}$ is then defined which includes a BCELoss term and any additional term which is trivially incorporated (e.g. the feature disruption term). We then iteratively perturb the source image for $K$ iterations using the sign of a collected momentum term. Similar to Dong et al. (2018) and Inkawhich et al. (2019), momentum is calculated as

$$m_{k+1} = m_k + \frac{\nabla_{I_k} L_{FDA}(I_k; \theta)}{||\nabla_{I_k} L_{FDA}(I_k; \theta)||_1},$$

where $m_0 = \mathbf{0}$ and $I_k$ is the perturbed source image at iteration $k$. The perturbation method for this $\ell_\infty$ constrained attack is then

$$I_{k+1} = Clip(I_k - \alpha * sign(m_{k+1}), 0, 1).$$

In this work, all attacks perturb for $K = 10$ iterations and $\alpha = \epsilon/K$.

## E. ADDITIONAL SALIENCY

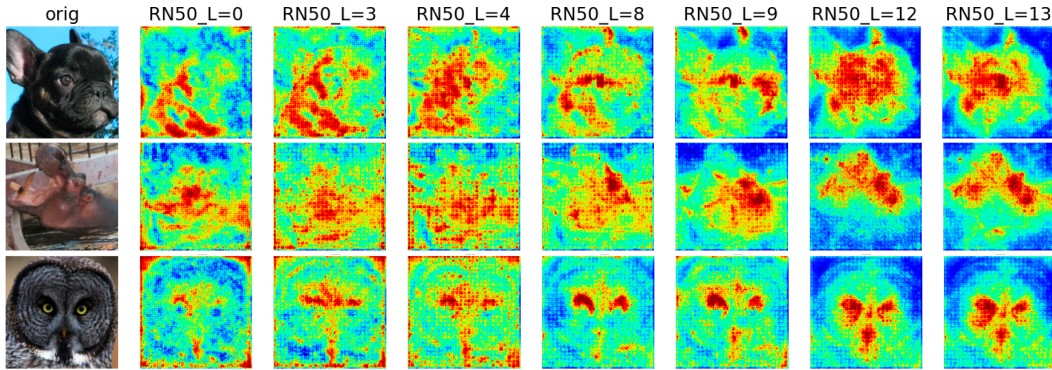

Figure 9: SmoothGrad saliency maps for RN50 auxiliary models.

