# OpenReview forum: "Transferable Perturbations of Deep Feature Distributions"
_ICLR.cc/2020/Conference — Accept (Poster)_

### Official Review · AnonReviewer2 · 2019-10-20
**Official Blind Review #2**

**Rating:** 8

**Review:**

The paper proposes a new adversarial attack for the targeted blackbox model Unlike previous approaches which use the output layer possibly with some additional terms and regularization, the proposed approaches only rely on intermediate features. In fact, the adversarial example is based on a single intermediate layer. The adversarial examples are built by training, for each target class, a binary classifier for the class based only on the features of that layer.

- There are multiple similar prior works that use intermediate layers in some way, and the paper does a good job to explain the differences between the proposed approach and these.

The paper proposes three variants. The simplest one appears rather weak in numerical experiments. The two other methods seem to achieve a different tradeoff, between focusing on the general error rate or on the targeted success rate.

In particular, FDA+fd is similar in some ways to Zhou et al. (2018) which uses an additional loss term to maximize the distance of the features at various layers. A comparison with this method (which uses the output layer) would be interesting, and may constitute a relevant additional baseline, at least for non-targeted metrics like error (which is where FDA+ms shines).

Similarly, a comparison to AA is provided in Figure 2 in the case of 10 classes, but not in Table 1 for the 1000 classes experiment. This casts some doubt on the significance of the result in the 1000 classes setting. A major issue with the proposed method is that training binary classifiers in large multi-class settings is quite costly, which is bound to hinder one's ability to identify 'high performing attack settings' In particular, the paper mentions that the 10 classes are used for that, but no evidence is provided that these settings are actually good for all classes.

The conclusions are rather consistent accross different pairs of source/target network.

The paper does not mention adversarial training. As this has become a common defence practice in the literature, it would be interesting to understand to what extent the proposed methods are thwarted by adversarial training. This could increase the significance of the paper if working with intermediate feature representation makes the attack more robust to this type of defense.

In summary, I find the experiments satisfying, although they can definitely be made more convincing by adding additional strong baselines and demonstrating how much performance can really be attained in the 1000 classes scenario.

- Generally, not using the output layer at all is an interesting approach that deserves in my opinion to be discussed and investigated further.

- The paper is clearly written, and notation is rigorous.

**Experience Assessment:**

I have read many papers in this area.

**Review Assessment: Checking Correctness Of Derivations And Theory:**

N/A

**Review Assessment: Checking Correctness Of Experiments:**

I assessed the sensibility of the experiments.

**Review Assessment: Thoroughness In Paper Reading:**

I read the paper at least twice and used my best judgement in assessing the paper.

---

> ### Author Response · Authors · 2019-11-10
> **Initial response to reviewer 2**
>
> Thank you for your thoughtful comments and thorough review. Here are a few points to address some of your comments:
>
> •	The reason for not including AA in Table 1 is because in Figure 6 of [1] the tSuc rate of the attack for ImageNet models is similar to the TMIM baseline. Thus, during evaluations we did not believe that it would add much information beyond what the TMIM baseline conveys. We can certainly try to fit it in the paper or appendix if you feel it is absolutely necessary.
>
> •	We agree that the influence of class choice on performance is an interesting topic, which we reserve for future work. Especially for ImageNet, it would be interesting to measure the transfer properties as we change the granularity of classes we are working with (e.g., attacking between terrier dog breeds versus attacking from a dog to a car).
>
> •	We also agree that understanding the implications of this method on Adversarially Trained (AT) models is an important future work, especially with recent observations of how AT models appear to learn different/more-robust feature sets [2]. However, given that there are currently no whitebox attacks that do well at attacking AT models, the potential efficacy of a blackbox transfer attack is unclear.
>
> [1] - Feature Space Perturbations Yield More Transferable Adversarial Examples
> [2] - Adversarial Examples Are Not Bugs, They Are Features

---

### Official Review · AnonReviewer3 · 2019-10-21
**Official Blind Review #3**

**Rating:** 3

**Review:**


The paper proposes a new transfer attack method (on undefended models), called FDA. The method uses auxiliary class-wise binary classifiers attached on intermediate layers, and optimizes the targeted probability of the binary model.

The proposed method is simple and shown effective according to the results in Table 1. But I have some serious concerns about the method and results.

Concerns
1. One weakness is the paper didn't seem to justify why using binary classification models on intermediate features is better than just using the final classification result in the original model. The authors should have an apple-to-apple comparison with optimizing the probability using the original model, including the "ms" variant, and also explain why using the auxiliary model is better than that. Even if you use the original model it is still utilizing the distribution of intermediate features, just through the main branch, so this reason alone does not seem convincing to me.

2. The baseline results seem too low (tpgd and tmim in Table 1, <10% target matching success rate). The FDA's result is ~20%. I'm not too familiar with the targeted adversarial attack literature, but one of the very early paper on targeted attack [1] seems to report much higher target matching rate (e.g., see their Table 3, ~70,80%). Why is this the case? What am I missing here? Is this just because they used an ensemble approach? If so you should also use ensemble to compare. Also the authors are welcome to want to point me to more recent papers' results and show their paper's result is better, given [1] is back in ICLR 2017.

3. One drawback of the method is it needs to train a custom model by oneself first, with one binary model for each class we want to attack or target. This can be very cumbersome in practice. For 1000 classes in ImageNet it is not possible to cover all of them. But other conventional methods don't need to worry about this, they just operate on a pre-trained model and supports any source and target class. I doubt whether the method is practically useful. This point should be acknowledged in the paper.

Overall, due to the concerns raised above (especially 2), I tend to vote a reject to the paper. I'm happy to reconsider my rating after the discussion period.

[1] Delving into transferable adversarial examples and black-box attacks.


**Experience Assessment:**

I have read many papers in this area.

**Review Assessment: Checking Correctness Of Derivations And Theory:**

N/A

**Review Assessment: Checking Correctness Of Experiments:**

I assessed the sensibility of the experiments.

**Review Assessment: Thoroughness In Paper Reading:**

N/A

---

> ### Author Response · Authors · 2019-11-10
> **Initial response to reviewer 3 - part 1**
>
> Thank you for your thoughtful comments and review. We now hope to address each of your concerns.
>
> 1)  In our design process, one of the goals was to investigate the transferability of perturbations of certain features, at the granularity of feature maps, and to decouple the attacking “signal” from the final layer information. This led to the idea of auxiliary models classifying intermediate feature maps. The auxiliary model technique was also appealing because it allowed for an interesting analysis of what is happening in each layer (e.g., Fig. 3). Also, [2] may explain the attack you are referring to, but they do not show targeted results (which may be a sign of poor performance given the implementation difference between targeted and untargeted attacks is a trivial sign flip between gradient ascent and descent on the loss signal).
>
> 2)  Here are some points we would like to clarify in the comparison of our work to [1]:
> •	Given that all of our results are from a single model transfer (to a *different* model architecture), and we only consider top-1 “matching rate,” we propose that the true apples-to-apples comparison results from [1] are the OFF-DIAGONAL numbers in Tables 2, 17, and 25. These tables all show black-box transfer at about 1-2% and at most 5% (when transferring within the ResNet family). Thus, these numbers are actually in line with our Table 1 baseline results.
> •	Table 3 of [1] (as directly mentioned in the review) shows the matching rate using an ensemble of 4 models and the optimization-based attack approach. After careful inspection, it appears that the off-diagonal values in this table may not be relevant for comparison as they are whitebox attack results. For an off-diagonal cell, the authors seem to have included the “blackbox” target model in the ensemble generating the attack. This is why the numbers are ~70,80%. Thus, this leaves only the on-diagonal values for relevant comparison. For these values, we see that when transferring to a model of the ResNet family the transfer rate is high because the majority of models in the ensemble are from the ResNet family. However, when the blackbox model is VGG-16 or GoogLeNet, the matching rate decreases to 24% and 11%. We propose that these are the most relevant numbers in this particular table for comparison (if we had to compare to an ensemble approach) because we do not consider transfers to models within the same family. However, this table used a 4-model ensemble to produce these results, whereas our evaluations only consider transfers from a single base model.
> •	Our method most closely relates to the gradient based approaches, rather than the optimization based approach.
> •	All results in [1] are measured over an evaluation set of only 100 images, and the target labels are chosen manually based on image semantics (page 4). This may lead to some variance in results when evaluating over such a small test set with hand-chosen target labels.
>
> Additionally, here are some relevant quotes from [1] (with care taken to preserve context) when the authors discuss transfers of targeted adversarial examples from a single model (as done in our work):
> •	 “We observe that … the targeted adversarial images can be rarely predicted as the target labels by a different model. We call the latter that the target labels do not transfer” (page 7)
> •	“Even if we compute the matching rate based on top-5 accuracy, the highest matching rate is only 10%” (page 7)
> •	“We also examine the targeted adversarial images generated by fast gradient-based approaches, and we observe that the target labels do not transfer as well” (page 7)
> •	“For targeted adversarial examples generated using FG and FGS based on an ensemble model, their transferability is no better than the ones generated using a single model. The results can be found in the appendix (Table 28, 29, 30 and 31)” (page 9)

---

> > ### Author Response · Authors · 2019-11-10
> > **Initial response to reviewer 3 - part 2**
> >
> > (Continues part 1)
> >
> > Also, as an extra precaution we have tested our implementation of the baseline attacks against the popular AdverTorch (https://github.com/BorealisAI/advertorch) library and found that our results match theirs. With this, we believe that the baseline results are low because the problem of transferable *targeted* attacks is indeed challenging.
> >
> > 3)  We agree that this is a cumbersome training process, which we do acknowledge in Section 4. However, in the manuscript we also mention that “an adversary would likely only be interested in attacking certain source-target pairs” which means they likely would not be interested in training all 1000. Also, once we found a highly-transferable layer, we *did* train all 1000 class models (Table 1), showing that it is at least possible. Another interesting thought is using multi-class auxiliary model classifiers. For example, if one was only interested in 5 classes, they may be able to train one 5-class auxiliary model per layer. This was considered in the design process, but in order to allow for maximum flexibility we opted for the binary classifiers. This is an interesting direction for future work. Finally, note that the base model is assumed to be pretrained.
> >
> > [1] - Delving Into Transferable Adversarial Examples and Black-box Attacks
> > [2] - Transferable Adversarial Perturbations

---

> > > ### Comment · AnonReviewer3 · 2019-11-14
> > > **Response**
> > >
> > > Thank you for the rebuttal. I'm satisfied with those part about [1] but I still think the proposed model's training process is too cumbersome to make it practically useful (training a model for each class), compared with other methods. Also, I'm not convinced by other reviewers' arguments for accepting the paper. That said, I have revised my rating to weak reject.

---

> > > > ### Author Response · Authors · 2019-11-14
> > > > **Additional comments on the training process**
> > > >
> > > > Thank you for the reply and the upgraded rating. We’re glad that we were able to clear up any confusion about [1] with respect to our results. Some additional comments about the training process:
> > > >
> > > > We want to emphasize that it is only necessary to train the auxiliary model $g_{l,c}$. Like other attack methods, we also used a pre-trained whitebox model $f$. When training $g_{l,c}$, we use $f_l$ ($f$ truncated at layer $l$) as the feature extractor, whose parameters remain frozen. Given that each $g_{l,c}$ is an MLP with two hidden layers of dimension 200, preparing the whitebox to generate attacks for a particular class at a certain layer is fast. Once $g_{l,c}$ has been trained, generating the corresponding adversarial examples is trivially quick, on par with the baseline methods.
> > > >
> > > > When we refer to the training process as being cumbersome, it is because we had 3 axes along which to experiment for the paper:
> > > > -	Whitebox-Blackbox combination: We tested 6 different whitebox-blackbox pairs (e.g. DN121->VGG19) to show a wide range of attack scenarios. In reality, the targeted blackbox is set, and from our experiments, it appears that DN121 is the best whitebox for transferable adversarial examples, so a real-world attack may only need to focus on one setting.
> > > > -	Classes: As previously mentioned, a real-world attacker is likely to be interested in attacking only a few (or one) classes, which further limits the number of auxiliary models $g_{l,c}$ needed, each of which in isolation is quick to train. If the attacker does want to target more, training $g_{l,c}$ is embarrassingly parallel, further reducing the training time needed.
> > > > -	Layer to attack: To produce Figure 2, we swept across various layers $l$ to identify the best layers for generating transferable examples. Our results in Figure 2 provide good guidelines to inform one’s search of $l$. One also does not need to train all 1000 classes while searching for $l$. We’ve found that using a subset (e.g. 10, as we did), generalizes well to the overall class set. Additionally, our results indicate that the best layer is relatively blackbox agnostic.
> > > >
> > > > To carry out a successful attack, a real-world attacker would only have to train a fraction of what we did. We carried out far more experiments as authors of a paper seeking to demonstrate the efficacy of our method. This work was done on a modest academic-lab sized hardware budget.

---

### Official Review · AnonReviewer1 · 2019-10-23
**Official Blind Review #1**

**Rating:** 8

**Review:**

This paper presents an adversarial attack based on the feature representations at different layers given the classes. Instead of only looking at the final layers, class samples in intermediate feature space information is used to attack and increase transferability at the same time. Then the noise is optimized to perturb the input so that the a specific wrong output will be more likely to chosen. The paper is clear and well-written and different interesting experiments support the claims.

1-	Intermediate feature space is not fully independent of the architecture of  the model. However, the results in Figure 2 shows schematically the same behavior for tSuc for different models. I am wondering how different the results would be if adversarial attack was trained on the black box model. In other words, as a simple testing for example, how much higher the success rate would be in the case of DN121->RN50 if you trained the noise on RN50 itself?
2-	For different scenarios explained in figure 2, the optimal layer is usually one of the intermediate ones, and it might change from case to case. To find the optimal layer, you needed to have access to the black box model. How possible do you think it is to make it black-box model agnostic? Maybe at the cost of less success?
3-	In the Figure captions you have “Figure” and in the text it is Fig. please make it consistent.


**Experience Assessment:**

I do not know much about this area.

**Review Assessment: Checking Correctness Of Derivations And Theory:**

I assessed the sensibility of the derivations and theory.

**Review Assessment: Checking Correctness Of Experiments:**

I did not assess the experiments.

**Review Assessment: Thoroughness In Paper Reading:**

I read the paper at least twice and used my best judgement in assessing the paper.

---

> ### Author Response · Authors · 2019-11-10
> **Initial response to reviewer 1**
>
> We appreciate your review and thoughts on the paper. Here are some responses to your comments:
>
> 1)  If we understand the question properly, “train[ing] the noise on RN50 itself” would constitute a whitebox attack of the RN50 where we can directly use the gradient signal of the model under attack. Whitebox attacks at this epsilon usually have ~100% success rate. Also, in this case the baseline methods represent some of the most powerful techniques in the whitebox setting because the noise is optimal for the model. On the other hand, our method is a blackbox attack where the gradient signal of the target model is unavailable in the noise generation process. This is one of the reasons that blackbox attacks (especially targeted ones) are so difficult.
>
> 2)  Finding the optimal transfer layer without experimenting with a blackbox model is a very interesting problem and arguably one of the most interesting opportunities for future work. If one is willing to train auxiliary models for the base model, they may get an intuition for high transferring layers by looking at the evolution of model saliency. Also, from Figure 2, the transfer properties of FDAs appear blackbox agnostic. Thus, if the attacker acquires/trains a pseudo-blackbox model in their “sandbox” environment, and only evaluates attacks against it, the good transferring layers observed in the sandbox setting are likely to be valid outside of the “sandbox” (to other unobserved blackbox models).
>
> 3) We will address this in the final version, thank you.

---

### Decision · Program_Chairs · 2019-12-19

**Decision:**

Accept (Poster)

**Comment:**

This paper considers black box adversarial attacks based on perturbations of the intermediate layers of a neural network classifier, obtained by training a binary classifier for each target class.

Reviewers were happy with the novelty of the approach as well as the presentation, described the presentation as rigorous and were pleased with the situation of this method relative to the literature. R3 had concerns about evaluation, success rate, and that the procedure was "cumbersome".  Some of their concerns were addressed in rebuttal, but remained steadfast that the method was too cumbersome to be practical.

I agree with R1 & R2 that this approach is novel and interesting and disagree with R3 that it is too impractical. The paper could be stronger with the addition of adversarial training experiments (and I disagree with the authors that "there are currently no whitebox attacks that do well at attacking AT models", this is very much not the case), but I concur with R1 & R2 that this is interesting work that may stimulate further exploration, enough so to warrant acceptance.